# Pembrolizumab Plus Axitinib for Metastatic Papillary and Chromophobe Renal Cell Carcinoma: NEMESIA (Non Clear MEtaStatic Renal Cell Carcinoma Pembrolizumab Axitinib) Study, a Subgroup Analysis of I-RARE Observational Study (Meet-URO 23a)

**DOI:** 10.3390/ijms24021096

**Published:** 2023-01-06

**Authors:** Marco Stellato, Sebastiano Buti, Marco Maruzzo, Melissa Bersanelli, Francesco Pierantoni, Ugo De Giorgi, Marilena Di Napoli, Roberto Iacovelli, Maria Giuseppa Vitale, Paola Ermacora, Andrea Malgeri, Brigida Anna Maiorano, Veronica Prati, Alessia Mennitto, Alessia Cavo, Matteo Santoni, Claudia Carella, Lucia Fratino, Giuseppe Procopio, Elena Verzoni, Daniele Santini

**Affiliations:** 1Medical Oncology Department, Fondazione IRCCS Istituto Nazionale dei Tumori di Milano, 20133 Milano, Italy; 2Department of Medicine and Surgery, University of Parma, 43121 Parma, Italy; 3Medical Oncology Unit, University Hospital of Parma, 43126 Parma, Italy; 4Medical Oncology Unit 1, Department of Oncology, Istituto Oncologico Veneto IOV-IRCCS, 35128 Padua, Italy; 5Medical Oncology Unit 3, Department of Oncology, Istituto Oncologico Veneto IOV-IRCCS, 35128 Padua, Italy; 6Department of Medical Oncology, IRCCS Istituto Romagnolo per lo Studio dei Tumori (IRST) Dino Amadori, 47014 Meldola, Italy; 7Department of Urology and Gynecology, Istituto Nazionale Tumori IRCCS Fondazione G. Pascale, 80131 Napoli, Italy; 8Department of Medical Oncology, Fondazione Policlinico Universitario A. Gemelli IRCCS Roma, 00168 Roma, Italy; 9Medical Oncology, University Hospital Modena, 41121 Modena, Italy; 10Department of Oncology, University and General Hospital, 33100 Udine, Italy; 11Department of Medical Oncology, Fondazione Policlinico Campus Bio-Medico, 00128 Roma, Italy; 12Oncology Unit, Foundation Casa Sollievo della Sofferenza IRCCS, 73013 San Giovanni Rotondo, Italy; 13Department of Medical Oncology, Ospedale Michele e Pietro Ferrero, Verduno—Azienda Sanitaria Locale CN2, Alba-Bra, 12060 Cuneo, Italy; 14SCDU Oncologia, “Maggiore della Carità” University Hospital, 28100 Novara, Italy; 15SSD Oncologia Ospedale Villa Scassi, ASL 3 Genovese, 16149 Genova, Italy; 16Oncology Unit, Macerata Hospital, 62100 Macerata, Italy; 17SSD Oncologia Medica, IRCCS Istituto Tumori Giovanni Paolo II, 70124 Bari, Italy; 18Division of Medical Oncology C, Centro di Riferimento Oncologico National Cancer Institute, 33081 Aviano, Italy; 19UOC of Medical Oncology, Sapienza Università di Roma, Polo Pontino, 00196 Latina, Italy

**Keywords:** non-clear renal cell carcinoma, cromophobe renal cell carcinoma, papillary renal cell carcinoma, pembrolizumab, axitinib, renal cell carcinoma, combination

## Abstract

Non-clear cell renal cell carcinoma (nccRCC) represents a heterogeneous histological group which is 20–25% of those with renal cell carcinoma (RCC). Patients with nccRCC have limited therapeutic options due to their exclusion from phase III randomized trials. The aim of the present study was to investigate the effectiveness and tolerability of pembrolizumabaxitinib combination in chromophobe and papillary metastatic RCC (mRCC) patients enrolled in the I-RARE (Italian Registry on rAre genitor-uRinary nEoplasms) observational ongoing study (Meet-URO 23). Baseline characteristics, objective response rate (ORR), disease control rate (DCR) and progression-free survival (PFS) and toxicities were retrospectively and prospectively collected from nccRCC patients treated in 14 Italian referral centers adhering to the Meet-Uro group, from December 2020 to April 2022. Only patients with chromophobe and papillary histology were considered eligible for the present pre-specified analysis. There were 32 eligible patients who received pembrolizumab-axitinib as first-line treatment, of whom 13 (40%) had chromophobe histology and 19 (60%) were classified as papillary RCC. The DCR was 78.1% whereas ORR was 43.7% (11 patients achieved stable disease and 14 patients obtained partial response: 9/19 papillary, 5/13 chromophobe). Six patients (18.7%) were primary refractory. Median PFS was 10.8 months (95%CI 1.7–11.5). Eleven patients (34.3%) interrupted the full treatment due to immune-related adverse events (irAEs): G3 hepatitis (n = 5), G3 hypophisitis (n = 1), G3 diarrhea (n = 1), G3 pancreatitis (n = 1), G3 asthenia (n = 1). Twelve patients (37.5%) temporarily interrupted axitinib only due to persistent G2 hand-foot syndrome or G2 hypertension. Pembrolizumab-axitinib combination could be an active and feasible first-line treatment option for patients with papillary or chromophobe mRCC.

## 1. Introduction

According to the World Health Organization (WHO) 2016 histological classification of renal tumors, non-clear cell renal cell carcinoma (nccRCC) represents a heterogeneous histological group, which is 20–25% of those with RCC [1]. Papillary and chromophobe RCC account for 80% of non-clear RCC and harbor histological, chromosomal alterations and molecular pathway different from clear cell RCC (ccRCC) and other nccRCC [2].

Papillary RCCs can be classified into two clinically and biologically distinct subtypes, type 1 RCCs associated to *MET* or *EGFR* alterations and type 2 RCCs with aggressive behavior and associated to *CDKN2A*, *SETD2* mutations [3]. Chromophobe RCCs are histologically similar to oncocytoma and frequently harbor mutations in *TP53*, *mTOR* and *PTEN* [4].

Patients with nccRCC have limited therapeutic options due to their exclusion from phase III randomized trials. Small prospective phase II trials investigated the role of everolimus and sunitinib in papillary and chromophobe RCC [5,6]. Crizotinib was shown to achieve an objective response and long-lasting disease control in patients with type 1 papillary RCC with MET mutations or amplification [7], whereas recently, cabozantinib demonstrated better activity in terms of the objective response rate (ORR) and progression-free survival (PFS) in papillary metastatic RCC (mRCC) compared to sunitinib [8].

The aim of the present study was to investigate the effectiveness and tolerability of the pembrolizumab-axitinib combination as a first-line treatment in chromophobe and papillary-metastatic RCC patients enrolled in the I-RARE study (Meet-URO 23).

## 2. Results

### 2.1. Overall Population

Thirty-two eligible patients received pembrolizumab-axitinib combination as first-line treatment in 14 centers adhering to the Meet-Uro network. Data collection was retrospective for 6/32 patients and prospective for 26/32 patients. Thirteen patients had chromophobe histology whereas 19 were classified as papillary RCC. The characteristics of the patients are reported in Table 1.

At a median follow-up of 7.1 months (95%CI 5.0–9.1), median PFS (mPFS) was 10.8 months (95%CI 7.8–13.7) (Figure 1).

ORR was 43.7% whereas the DCR was 78.1% (11 patients achieved SD and 14 patients obtained PR as best response). Six patients (18.7%) were primary refractory (Table 2).

The median duration of treatment was 7.5 months (95%CI 6.3–8.7).

The median OS was not reached and 94.6% of patients were alive at 1 year.

Eleven patients (34.3%) interrupted the full treatment due to immune-related adverse events (irAEs): G3 hepatitis (n = 5), G3 hypophisitis (n = 1), G3 pancreatitis (n = 1), G3 diarrhea (n = 3), 1 asthenia G3; 12 (37.5%) patients interrupted axitinib due to persistent G2 hand-foot syndrome or G2 hypertension.

### 2.2. Chromophobe Renal Cell Carcinoma

In chromophobe RCC patients, mPFS was not reached. Those experiencing disease progression at one year were 4/13 (30.7%). ORR was 41.6% whereas the DCR was 76.9% (five patients achieved PR and five patients achieved SD) (Figure 2 and Figure 3). Two patients were primary refractory (Table 2). One patient was not evaluable due to rapid clinical deterioration.

### 2.3. Papillary Renal Cell Carcinoma

In papillary RCC patients, mPFS was 10.8 months (95% CI4.48–17.12, Figure 4). ORR was 47.3% whereas the DCR was 78.9% (9 patients had PR whereas 6 patients had SD) (Figure 2 and Figure 3). Four patients had progressive disease (PD) as best response (Table 2).

## 3. Discussion

Chromophobe and papillary renal cell carcinoma are a rare subtype of RCC, classified as nccRCC, that account for 5% and 10–15% of RCC cases, respectively [9,10]. Favorable prognosis is attributed to papillary type 1 and chromophobe-localized RCC whose cancer specific survivals at 5 years are 87.4% and 86.7% compared to 68.9% for ccRCC [11]. Otherwise, papillary type 2 is more aggressive and tend to rapid metastatic spread [12].

The choice of therapy for nccRCC is often a challenge and limited evidence is available to guide clinicians in treatment, due to its relative rarity. Although immune-checkpoint inhibitors (ICI) based combinations can be considered as standard of care for metastatic RCC [13], pivotal trials included only patents with ccRCC, therefore nccRCC have limited personalized treatments [14,15].

Recently, Procopio et al. demonstrated the efficacy of cabozantinib in metastatic collecting duct carcinoma (CDC) [16]. The study prospectively enrolled patients for central pathology review and was a breakthrough for the treatment of nccRCC considering the rarity of disease and that the only prospective trial for CDC was dated 2007 and included patients treated with platinum-based chemotherapy [16].

Pale et al., in a randomized phase II trial including only patients with papillary mRCC, reported a significant improvement in both PFS and response rate with the dual VEGF/MET inhibitor cabozantinib compared to sunitinib (9.0 months (95% CI6–12) vs. 5.6 months (95%CI 3–7), respectively) [8]. For papillary and chromophobe RCC, ESMO, and NCCN, guidelines still suggest tyrosine-kinase-inhibitors (TKI) monotherapy or everolimus, but the results of these approaches demonstrate that more effective treatment options for nccRCC are needed. Indeed, enrollment in clinical trials is recommended [17,18].

To date, pembrolizumab-axitinib combination has been demonstrated to prolong PFS, OS, and ORR in metastatic ccRCC compared to sunitinib. Nevertheless, papillary and chromophobe RCC represent different renal malignancy and treatments should not be habitually extrapolated from RCC. There are biological differences between clear and non-clear RCC that raise some doubts about transposing our knowledge for ccRCC to nccRCC. Nevertheless, pembrolizumab monotherapy showed an attractive ORR in papillary and chromophobe RCC (28.8% and 9.5%, respectively) and 4.2 months (95% CI 2.9–5.6) as PFS in a phase II prospective trial [19].

Lately, Graham et al. reported the outcome of nccRCC patients treated with IO, reporting improvement in OS compared to TKI monotherapy. Despite 49 papillary and 28 chromophobe patients being treated with the ICI-based combination, only 14 patients were treated with the IO-TKI combination [15] with limited data supporting this association.

Despite the rarity of these subtypes of RCC, we collected 32 patients treated with pembrolizumab-axitinib combination with a median follow-up of 7.1 months, which is consistent to previous reports in literature. In our cohort of previously untreated patients with chromophobe or papillary mRCC, pembrolizumab-axitinib combination showed therapeutic potential with an overall ORR of 43.7%, DCR of 78.1% and mPFS of 10.8 months. The subgroup analysis for different histology showed promising ORR and DCR in both groups, 47.3% and 78.9% in papillary patients and 41.6% and 76.9% for chromophobe patients, respectively.

The safety profile of pembrolizumab-axitinib combination in the present study seems consistent with what has been observed in the KEYNOTE-426 trial [20] that showed in the experimental arm a discontinuation rate of either drugs in 30.5% of patients and reported arterial hypertension as the most common AE.

The limits of our trial include the short follow-up, a lack of central radiological and histological revision, and the partially retrospective collection of data. Indeed, an ambispective trial is not ideal to study the effectiveness and tolerability of IO-combination because some data have been collected retrospectively, within the limit of a retrospective collection.

Based on our results, pembrolizumab-axitinib combination could be considered a potential option for chromophobe and papillary RCC patients.

## 4. Materials and Methods

Baseline characteristics, ORR, disease control rate (DCR), PFS, and toxicities were retrospectively and prospectively collected from nccRCC patients treated in 14 Italian referral centers adhering to the Meet-Uro group, from December 2020 to April 2022. Only patients with chromophobe or papillary histology treated with the pembrolizumab-axitinib combination as first line were considered eligible for the present analysis. Data were extracted from patients enrolled in the I-RARE study (Meet-URO 23).

Meet-URO23 (I-RARE study) is a multicentre trial aimed to collect data about rare genitourinary cancers. The registry is an ambispective observational real-world collection of patient characteristics, treatment and outcome. The study was approved by the ethics committee of Istituto Oncologico Veneto, Padua, Italy (No. 2021/19/PU).

Written informed consent was provided by all the patients or a legally authorized representative. All participating centers received local ethics approval for data collections. The study was conducted in accordance with good clinical practice and the Declaration of Helsinki.

Considering the explorative intent of this observational study a sample size calculation was not provided.

Primary endpoint was the PFS, defined as the time from the start of pembrolizumab-axitinib combination to radiological or clinical progression or death, whichever occurred first. Real-world physician-assessed progression and response were based on radiographic criteria using Response Evaluation Criteria in Solid Tumors (RECIST) guidelines [21], with imaging assessments occurring about every three months. In case of missing radiological assessment due to rapid clinical deterioration, clinical criteria were used to define progression. PFS is considered as an important measure of treatment benefit and can be evaluated earlier with fewer patients and no confounding data due to subsequent treatment.

Secondary endpoints were ORR, DCR, overall survival (OS) and toxicity. ORR included partial and complete responses (CR+PR), whereas DCR includes ORR plus stable diseases (SD) as best responses. OS was calculated from the pembrolizumab combination start to death for any cause. Toxicities were measured according to the Common Terminology Criteria for Adverse Events, version 5.0 (CTCAE v 5.0)

Patients without progression or death were considered censored at the date of last follow-up.

Baseline demographic and clinical characteristics have been described using frequencies and percentages for categorical variables. Descriptive analysis was made using median values and ranges. Survival curves were built by the Kaplan–Meier method. All statistical analyses were performed using SPSS software (version 19.00, SPSS, Chicago, IL, USA). The median follow-up was calculated using the reverse Kaplan–Meier method [22].

## 5. Conclusions

Pembrolizumab-axitinib combination is an effective option for chromophobe and papillary RCC but further studies, including only these subtypes of RCC, are needed to prospectively confirm the efficacy of novel treatments with combinations for non-clear RCC.

## Figures and Tables

**Figure 1 ijms-24-01096-f001:**
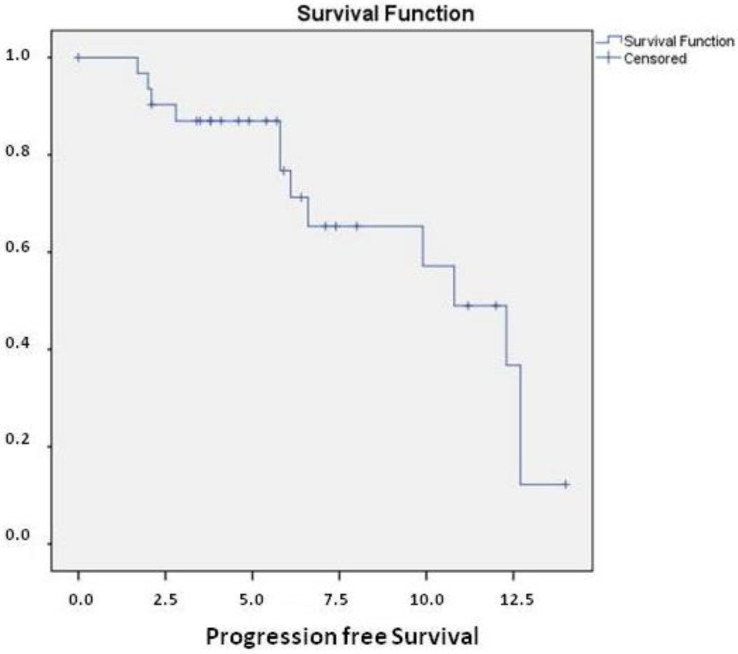
median Progression Free Survival (mPFS) in the overall population was 10.8 months (95%CI 7.8–13.7).

**Figure 2 ijms-24-01096-f002:**
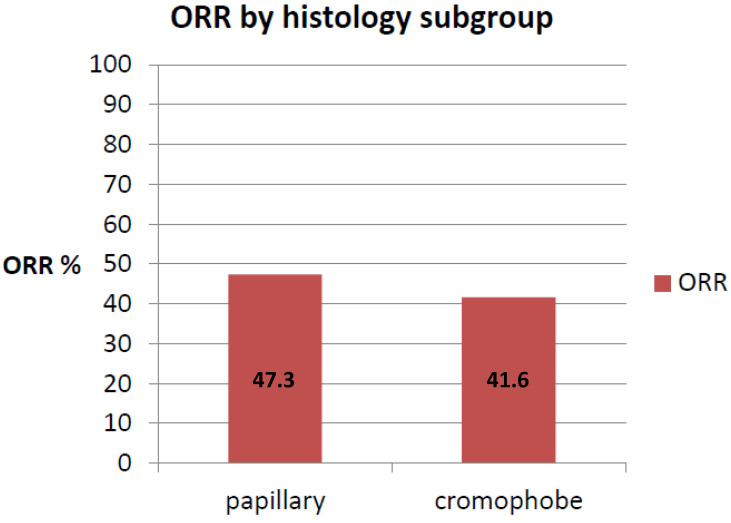
ORR according to histology subgroup.

**Figure 3 ijms-24-01096-f003:**
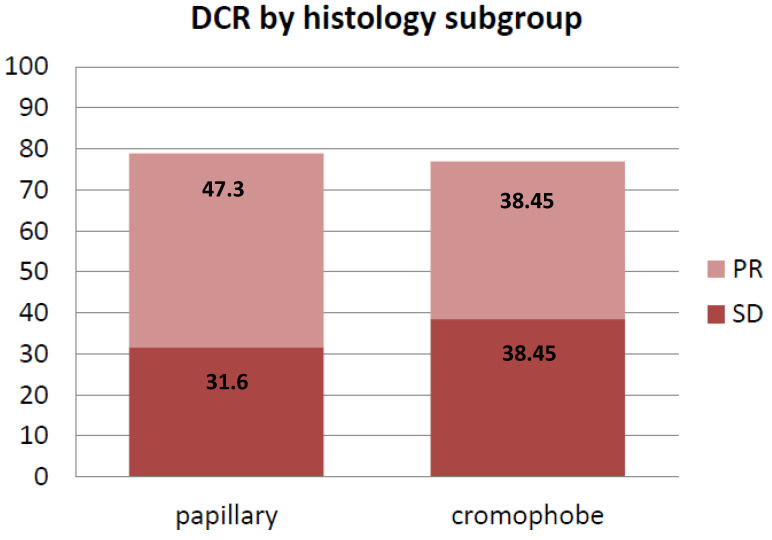
DCR according to histology subgroup.

**Figure 4 ijms-24-01096-f004:**
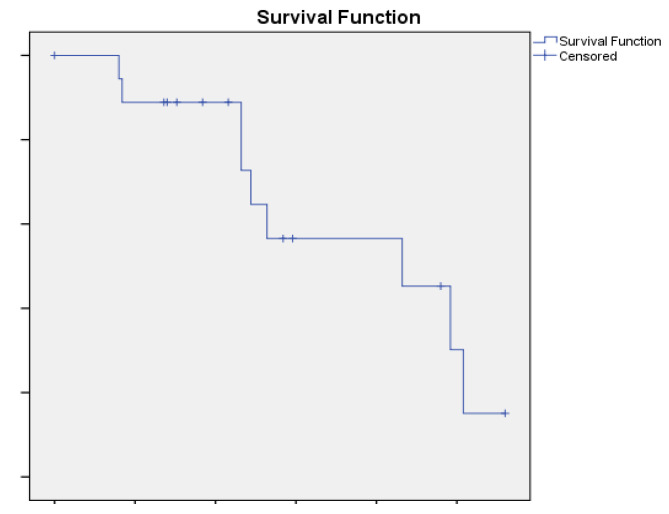
median Progression Free Survival (mPFS) in the papillary RCC patients was 10.8 months (95%CI 4.48–17.12).

**Table 1 ijms-24-01096-t001:** Characteristics of patients. IMDC (International Metastatic RCC Database Consortium); ECOG PS (Eastern Cooperative Oncology Group Performance Status); yrs (years); met (metastasis).

	N (%)
Median Age	68 yrs
M	23/32 (72)
F	9/32 (28)
Chromophobe RCC	13/32 (40)
Papillary RCC	19/32 (60)
Sarcomatoid features	3/32 (9)
IMDC score	
good	4/32 (12)
intermediate	20/32 (62)
Poor	8/32 (25)
Previous Nephrectomy	23/32 (72)
Cytoreductive Nephrectomy	17/32 (53)
ECOG PS	
0	21/32 (65)
1	7/32 (22)
2	5/32 (13)
Synchronous metastatic disease	13/32 (40)
Bone met	4/32 (12)
Liver met	13/32 (40)
Lung met	8/32 (25)
Nodes met	24/32 (75)

**Table 2 ijms-24-01096-t002:** Radiological Response in overall population and according to different histologies.

	Overall Population (n = 32)	Chromophobe Histology (n = 13)	Papillary Histology (n = 19)
CR	0	0	0
PR	14 (43.8)	5 (16.1)	9 (19.3)
SD	11 (34.3)	5 (19.3)	6 (16.1)
PD	6 (18.8)	2 (12.9)	4 (6.4)
NE	1 (3.1)	1 (3.1)	
DCR	78.1%	78.9%	76.9%
ORR	43.7%	41.6%	47.3%

CR (Complete Response); PR (Partial Response); SD (Stable Disease); PD (Progressive Disease); NE (Not Evaluable); DCR (Disease Control Rate); ORR (Objective Response Rate).

## Data Availability

Data are available upon request.

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
