# Peer review of "Pembrolizumab Plus Axitinib for Metastatic Papillary and Chromophobe Renal Cell Carcinoma: NEMESIA (Non Clear MEtaStatic Renal Cell Carcinoma Pembrolizumab Axitinib) Study, a Subgroup Analysis of I-RARE Observational Study (Meet-URO 23a)"

_ijms, 2023, doi:10.3390/ijms24021096_

Round 1
Reviewer 1 Report
This is an interesting real-world, multi-institutional, academic, and non-academic center experience in non-clear cell renal cell carcinoma (nccRCC) which comprise several rare diseases, often characterized by bad prognosis, with no standard treatments available, and under-represented in prospective randomized trials. The paper is well-written and carefully presented.
Below are some major concerns:
1. Design of the study: this is an “ambispective observational real-world collection” from the I-Rare Trials. Although an ambispective trial is not ideal to study the effectiveness and tolerability of pembro-axitinib combination (the best study should have been a prospective study), the authors should mention the number of patients whose data were collected prospectively and prospectively. Also, the methodological limitations should be discussed more in the discussion.
2. Endpoints: most of the studies which investigate the effectiveness and tolerability of a single treatment or combination of treatments in a single arm trial in nccRCC had as a primary endpoint the ORR. The authors should justify why they have chosen the PFS as the primary endpoint and not the ORR. See some similar studies below:
https://pubmed.ncbi.nlm.nih.gov/35420628/
https://pubmed.ncbi.nlm.nih.gov/35442713/
https://pubmed.ncbi.nlm.nih.gov/33529058/
https://pubmed.ncbi.nlm.nih.gov/31721643/
3. If the primary endpoint is the PFS, the authors should provide the Kaplan-Mayer figure with all the details.
4. The authors should provide more information about the PD-L1 expression, and the outcome based on the PD-L1 expression.
5. 23 (72%) of treated patients temporarily interrupted or interrupted the treatment due to irAEs. Despite these safety data, the authors state that pembro and axitinib is an effective treatment. Because the aim of the study was also the tolerability of the drug combination, the authors should discuss more the data about safety. Also, the authors state “The safety profile of pembrolizumab-axitinib combination in the present stud seems consistent with what has been observed in KEYNOTE-426 trial22.” Could they please explain more these sentences?
6. The limitations should be discussed more: the ambispective design, the lack of histopathological, radiological revision and PD-L1 expression, the number of patients, the length of follow-up. Also, some data are collected retrospectively, therefore the authors relied on the clinical information reported in the charts for the best response and irAes and thus some irAEs may have been misclassified or not reported.
Minor concerns:
1. Check if the data reported in the results are correct: “At a median follow-up of 7.1 months (95%CI 5.0-9.1), median PFS (mPFS) was 10.8 months (95% CI 7.8-13.7).”
Author Response
I would like to really thank both reviewers for the suggestions that improved our paper. I hope that with these revisions, the article would be ready for pubblication.
- In the results and the discussion sections we mentioned for how many patients we collected data retrospectively and prospectively. Only for 6/32 patient the collection was retrospective, with limited bias due to the known problems of a retrospective collection. Furthermore, in the discussion we discussed more deeply the limits of our trial as suggested in point 6, including the ambispective design and the lenght of follow up. We did not include the lack of PDL-1 analysis as a limit beacause in mRCC, the role of PDL-1 is still controversial and it is not commonly use in clinical practice. The prognostic and predictive role of PDL-1 merit furthereconsiderations but not in this contest. The number of patients is small, but considering the rarity of the histologies included in our trial, it is adeguate to support the results. A prospective and larger trial is desirable but difficult to explore in a real world collection.
- PFS is considered as an important measure of treatment benefit and can be evaluated earlier with fewer patients and no confounding data due to subsequent treatment. Association between ORR and OS are opinable. The suggested papers are inspiring and foundamental to support the efficacy of treatments in selected category of patients. BONSAI trial was conducted by our group and ORR was used to support the efficacy of cabozantinib, with prospective enrollent and withour confounding bias due to retrospective collection of data. ORR could be difficult to be registered in retrospective collection and can be limited by clinical criteria used sometimes to define progression.
- Kaplan-Mayer have been provided
- As reported in point 1
- We explain more deeply the comparation of toxicities with KEYNOTE-426 in which a discontinuation rate of either drugs was reported in 30.5% of patients and arterial hypertension was the most common AE, similarly to our report.
- As reported in point 1
Regarding minor concerns, the median follow-up was calculated using reverse Kaplan–Meier method.
Reviewer 2 Report
In the Method, PFS was estimated by K-M curve, which is not shown in the Result.
The discussion is poorly structured, consisting of many paragraphs with only one sentence.
The results should be compared to Keynote 426 in a more detailed fashion.
Author Response
I would like to really thank both reviewers for the suggestions that improved our paper. I hope that with these revisions, the article would be ready for pubblication.
- K.M curves have been added to the paper
- The discussion have been modified in a more elegant fashion with more structured sentences
- In the discussion, the results have been compared to KN-426 that showed in the experimental arm a discontinuation rate of either drugs in 30.5% of patients and reported arterial hypertension as the most common AE.
Round 2
Reviewer 1 Report
The authors clearly answered my comments and questions.